# Investigation on Micro-Hardness, Surface Roughness and SEM of Nano TiO₂/B₄C/Graphene Reinforced AA 7075 Composites Fabricated by Frictional Stir Processing

**Majed Mohammed Hotami [1,2,\*] and Shengyuan Yang [1]**

[1] Department of Mechanical and Civil Engineering, Florida Institute of Technology, 150 West University Boulevard, Melbourne, FL 32901, USA; shengyuanyang@gmail.com
[2] Department of Mechanical Engineering, Umm Al-Qura University, Makkah City 24372, Saudi Arabia
[\*] Correspondence: mhotami2015@my.fit.edu

**Abstract:** The current work seeks to discover and choose the proper friction stir processing (FSP) settings for aluminum alloy 7075 surface composites enhanced by adding three unique nanoparticles of titanium dioxide (TiO₂), B₄C, and graphene for superior performance. FSP is the only method that produces higher amounts of particle distribution and nanoscale reinforcing. For the sample fabrication, a special relatively high rotational speed of 2000 rpm and feed rate of 45 mm/min were tested with a suitable range of processing parameters (800–2000 rpm, 25–45 mm/min). To measure the micro-hardness and surface roughness of three different surface nano composites, they were studied under various FSP conditions. The findings showed that surface composites produced at high rotational speeds of 1400 rpm and 45 mm/min decreased surface roughness and granule distributions by 39% and 73%, respectively, and increased surface micro-hardness by 54%. According to the microstructure investigations, good bonding was produced between the AA7075 substrate generated at 1200 rpm and the base metal and friction stir processed specimens at 800 and 2000 rpm. The AA7075/B₄C surface composite produced at 1200 rpm rotating speed had a higher micro-hardness than the other two surface composites.

**Keywords:** friction stir processing; surface nanocomposites; micro-hardness; AA7075/TiO₂; B₄C/graphene





## 1. Introduction

Surface properties, in particular a combination of superior bulk material toughness and strong surface wear resistance, govern the durability of parts in a number of applications. For this reason, it is desirable to include different nanoparticles into component surfaces to boost their tensile strength while preserving the bulk material's composition and structure. Several solid-state methods are used to effectively and uniformly disperse particles in the matrix material. Surface engineering is a fundamental engineering strategy that is used for a range of applications because it has produced significant energy savings, particularly for hardness, increasing surface, and tribological applications [1]. In a variety of tribological, mechanical applications, surface features are more important than bulk material qualities [2]. Therefore, surface features of the substances are frequently emphasized in tribological applications without much attention paid to scanning electron microscopy (SEM), hardness, or surface roughness parameters. Surface composites can enhance surface qualities while preserving the benefits of the primary material [3]. Friction stir processing (FSP), a method for surface engineering, may be utilized to create surface composites while being energy- and eco-friendly [4]. Surface composites may be manufactured using the groove, cover-plate, and drill-hole FSP methods [5]. FSP is used to create surface composites employing the friction stir tool and the capping tool, two different types of equipment. This equipment uses a capping tool to seal the reinforcements and a friction stir tool to combine the metal matrix on the surface with the reinforcing particles [6].

Micro-sized ceramic particle reinforced AMCs have a restricted use due to their poor malleability and low fracture toughness. Using nanosized particles, the AMC's hardness, surface roughness, and ductility are improved, as opposed to friction stir processing (performed in the air), which results in AA6061 with exceptionally small grains of 200 nm [7]. Tool geometry and FSP parameters' influence the microstructure of the AS-cast alloy and Al-Si-Mg alloy. A high tool rotation rate is the most efficient parameter for changing grain size and thus, boosting strength [8]. When FSPed and cold-rolled AA5083 specimens were compared, the super plasticity was improved by stirring at a transverse speed and with the least amount of tool rotation. It also increased elongations, lowered flow stresses, and enhanced the grain size of the recrystallized material [9]. A square pin profile generated an area of the FSP without any faults, according to results of the FSP zone formation in AA2219. Additionally, joints with a 1600 rpm rotational speed demonstrated better tensile characteristics [10]. The $Al_2O_3$ nanoparticles for the AA2024 matrix alloy's wear mechanism showed that the increase in rotational speed (from 640 to 800 rpm) had some impact on the microstructure's grain size [11]. A surface composite with a smaller grain size than FSPed of base material was created by FSP of multi-walled carbon nanotubes (CNT) in the AZ31 magnesium alloy, but the micro-hardness increased by two times [12]. The 1.8 nm surface area of a cast plate was changed into tiny (1.2 nm diameter) equiaxed fundamental grains with increased fatigue life, according to an investment cast and HIP Ti-6Al-4V plate with altered microstructures created by the FSP [13]. Super austenitic steel's fundamental structure has a hardness of 185 HV; however, nano-structured super austenitic steels produced by FSP showed a considerable improvement in hardness up to 350 HV [14]. Surface composite layer hardness over base material increased by a factor of two when an FSPed AA/$Al_2O_3$ composite surface layer was filled in a shallow groove and subjected to wear behavior. Moreover, it showed enhanced wear resistance [15].

FSP's impact on the microstructure of AL6061 sheets transformed their coarse grain into very fine ones, increasing their hardness by 50%. Highly pressurized torsion increased the AA's yield strength and hardness, enabling a high strain rate [16]. To produce hierarchical metal matrix composites by surface modification, graded structures, stir zone of FSPed AA5083 Al powders with $B_4C$ particles atomized by inert gas (SZ) demonstrated enhanced ductility and strength [17]. The surface of the AA2009 FSPed with rolled composite material samples reinforced with 1.5–4.5 vol.% CNT were manufactured at a tool rotation rate of 1200 rpm and a travel speed of 100 mm/min indicated that 3 vol.% of CNT had maximum tensile strength and substantially greater elongation [18]. AZ31/TiC magnesium metal matrix composites exhibited TiC particles adequately bonded to the magnesium matrix with no interfacial interaction between the matrix and the reinforcement using a single pass FSP at a tool rotation speed of 1200 rpm, a traverse speed of 40 mm/min, and an axial force of 10 kN [19].

After the inclusion of micro- and nano-boron carbide particles to the composites, it was discovered that the nanocomposites had higher tensile strength, ductility, and impact energy compared to those of the micro $B_4C$ nanoparticles [20]. FSPed AA5052/$TiO_2$ ultra-fine-grained nanocomposites with improved tensile yield strength and micro-hardness were discovered using nanoparticles under freezing conditions. Producing surface composites with the required performance depends on the rotation speed and feed rates, which control a considerable amount of heat production during FSP [21]. Titanium dioxide ($TiO_2$)-supplemented FSPed AA 6063 surface composites have tailored surface micro-hardness. As a reliable, eco-friendly component, $TiO_2$ has been employed in many industries [22]. In Cu/$TiO_2$ FSPed composites, $TiO_2$ enhanced the substrate material's surface hardness, yield strength, and micro-hardness. Because of its ease and ability to make high-performance CNT macro-assemblies with limitless size and desired morphologies, the floating catalyst technique has gained industrial success. The factors of synthesis, material qualities, post-treatment techniques, and their potential uses are discussed [23].

The $TiO_2$ nano-powder improves the micro-hardness, yield strength, and tensile strength of the AA 5052 substrate material. Fullerenes, CNTs, graphene (Gr), and nano-diamonds are discussed in detail, as well as the characteristics of CNT and graphene that make them

exceptionally lubricious [4]. Wet-spinning and dry-spinning techniques can be used to create CNT strands with infinite length, regulated morphologies, and aligned structures. Post-treatments, such as densification, acidification, and hybridization, can enhance electrical and mechanical characteristics, implying that they have a wide range of uses [24].

There are few studies on the production of AA7075/TiO$_2$/B$_4$C/Gr surface composites, and metallurgical, mechanical, and FSPed composite qualities have not been discussed. Utilizing a wide range of rotational speeds (800, 1000, 1200, 1400, and 2000 rpm) and a transverse speed of 25–45 mm/min, fabricated surface composites were examined for their microstructure, as well as their micro-hardness and surface roughness features that improve other properties, such as tribology and corrosion. The goal of the current investigation was to fabricate AA 7075/TiO$_2$/B$_4$C/Gr and to improve the surface performance of AA7075 substrate material using suitable FSP fabrication parameters. This work seeks to investigate the effects of three distinct reinforcement nanoparticles from various families of the FSPed nanocomposite.

## 2. Materials and Methods

### 2.1. Material Matrix and Reinforcement Particles

The heat treatable 7xxx series, whose main ingredient is Al-Zn-Mg, comprises AA7075; the chemical composition of AA7075 is shown in Table 1. Thick AA7075 plates (6 mm) were obtained and sliced into 150 mm × 50 mm × 6 mm specimen sizes for processing. TiO$_2$, B$_4$C, and graphene powders with average particle sizes of 36.85 nm, 900 nm, and 10–15 nm, respectively, were purchased from Nano Research Lab in India.

**Table 1.** Chemical composition of AA 7075.

| Element | Zn | Mg | Cu | Si | Fe | Ti | Cr | Al |
|---|---|---|---|---|---|---|---|---|
| **Amount (%)** | 5.9 | 2.3 | 1.3 | 0.2 | 0.3 | 0.08 | 0.04 | Balance |

### 2.2. Groove Cutting and Reinforcement Particles Allocation

In this work, the most common approach for creating composite aluminum surfaces—groove filling FSP—was used. A groove with the dimensions (175 mm length, 3 mm width, and 2 mm depth) was created using a milling machine. The masses of each TiO$_2$ nano-powder (0.2 g), B$_4$C (0.2 g), and Gr (0.2 g) were determined using an electronics balance to ensure that each nano-powder integration into each workpiece was uniform. Figure 1a–c shows the FSP machine, FSP tool, and the placement of TiO$_2$, or B$_4$C, or Gr nano-powder in their groove on three different AA 7075 substrate materials.

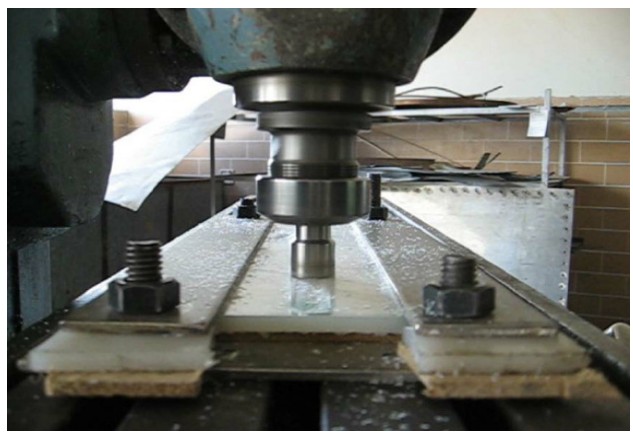

(a)

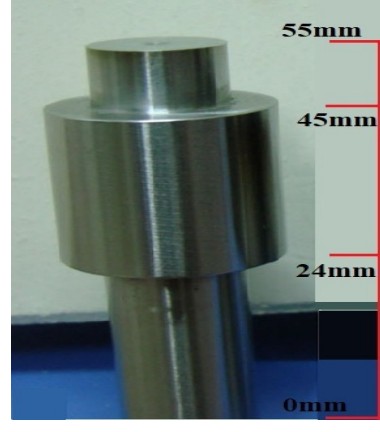

(b)

**Figure 1.** *Cont.*

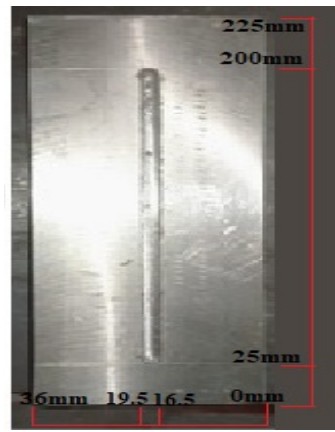

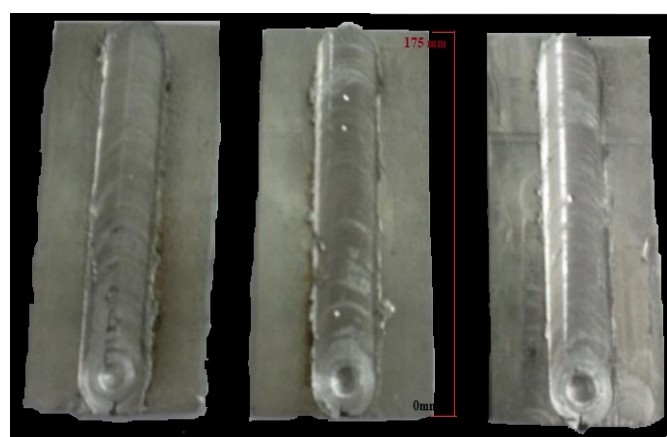

(**c**)                              (**d**)

**Figure 1.** (**a**) FSP machine (**b**) FSP tool (**c**) groove formed (**d**) B4C/TiO$_2$/Gr inserted in the groove on the substrate material.

### 2.3. Capping Process

The capping technique was performed using a pinless FSP tool attached to a milling machine. Using a rotating tool, the grooves were sealed with TiO$_2$, B$_4$C, and Gr nano-powders. The depth of the capping was 0.3 mm. The pinless FSP tool was constructed from ASP 23 high-speed steel. An 800-rpm tool rotation rate, 30 mm/min tool traverse speed, and 0° tool tilt angle were the processing conditions for the capping pass. In Table 2, the seven different tool rotation rates (800, 1000, 1200, 1400, and 2000 rpm) and traverse speed (25–45 mm/min) for the stirring processes are provided. The purpose of the groove in the plates was to improve powder dispersion and reduce the amount of powder that accumulated on the advancing side during FSP.

**Table 2.** FSP parameters used for the fabrication of FSPed AA 7075/TiO$_2$/B$_4$C/Gr surface composite specimens.

| Samples Numbers from 1 to 36 | | | Processing Parameters | |
| --- | --- | --- | --- | --- |
| AA7075/TiO$_2$ | AA7075/B$_4$C | AA7075/Gr | Rotational Speed (rpm) | Feed Rate (mm/min) |
| 1 | 13 | 25 | 800 | 25 |
| 2 | 14 | 26 | 800 | 35 |
| 3 | 15 | 27 | 800 | 45 |
| 4 | 16 | 28 | 1000 | 25 |
| 5 | 17 | 29 | 1000 | 35 |
| 6 | 18 | 30 | 1000 | 45 |
| 7 | 19 | 31 | 1400 | 25 |
| 8 | 20 | 32 | 1400 | 35 |
| 9 | 21 | 33 | 1400 | 45 |
| 10 | 22 | 34 | 2000 | 25 |
| 11 | 23 | 35 | 2000 | 35 |
| 12 | 24 | 36 | 2000 | 45 |

### 2.4. Microstructure Studies

For both the friction stir-treated base metals and the specimens, specimens were sectioned perpendicular to the processing direction. Using a disk grinder, samples were planar ground at different grits, from coarse to fine. These grits included 300, 600, 800, 1000,

1200, and 2000 mm. Diamond paste with a 3 μm thickness was used for rough polishing on velvet pads. With a 0.05 μm Alumina solution, fine polishing was carried out with micro-cotton pads. Using Keller's reagent, the sample specimens were chemically etched. The samples were extensively cleaned to remove the carbon deposits, then dried for more microstructural research. The results of microstructural testing on the FSPed specimens containing reinforcing particles were obtained using a scanning electron microscope (SEM) (VEGA 3 TESCAN, Brno—Kohoutovice, Czech Republic).

### 2.5. Micro-hardness of Surface Composites

According to the ASTM E384 standard and Vickers hardness tester, micro-hardness testing determines the impact of plastic deformation and grain reorientation on the mechanical characteristics of the weld nugget on test samples. Using a test load of 9.81 N and a dwell length of 10 s using a square-based pyramid-shaped diamond indenter, the micro-hardness indentations of the recorded measurements were taken as near as feasible to the center of the thickness and then measured and translated into hardness values. By ensuring that the specimen is maintained perpendicular to the indenter to enable accurate measurements, the specimens' smooth, polished surface produces a consistently shaped indentation. To make testing easier, the specimens were placed on a plastic medium. The samples were polished and ground before the micro-hardness was determined. Six indentations were produced along the width of the stir zone formed on the top of the FSP surface of the specimen, and the locations of each indentation were noted.

### 2.6. Surface Roughness (Ra, Rq and Rz) Evaluation

By using a Taylor Hubson Pro-filometer, the surface roughness parameters Ra (the average roughness), Rz (the difference between the peaks and the valleys), and Rq (the root mean square) created by different FSP processing factors were measured. Surface roughness was studied in relation to rotational speeds and feed rates.

### 3. Results and Discussion

### 3.1. Results of Microstructure Analysis

FSPed AA 7075 is depicted in Figure 2a,b as having an alpha solid solution dendritic matrix in white and a second phase zinc-rich eutectic coarse structure in black. A fine and equal axis grain structure indicative of the stir region was observed in the treated specimens, which was caused by plastic/permanent deformation and significant heat produced from friction (refer to Figure 3). Figures 4 and 5 show the metallographs of FSPed specimens with TiO$_2$ at various rotating speeds and feed rates. The FSPed specimen at 1400 rpm showed better particle distribution in the SEM analysis, as evidenced by the absence of TiO$_2$ nanoparticle assemblage, and appears to have finer grain size refinement, which is brought on by the generation of high heat energy and the availability of time for the substance to flow along the substrate [1]. It has been established that these effects are caused by high rotation speed because it alters grain size and generates new nucleation sites. Because the process happens fast, feed rates are adjusted at different rotational speeds to prevent a diminished heat input. To avoid the pinning effect and aggregation of the particles, constant traversal speed is advised. Increasing the tool rotational speeds or the rotational rate/traverse speed ratio will erase the banded structure. Uniform dispersion of nanoparticles is the result of a higher stirred zone temperature and, consequently, better material mixing. This is due to the amplified heating and stirring caused by friction when a tool rotates more quickly.

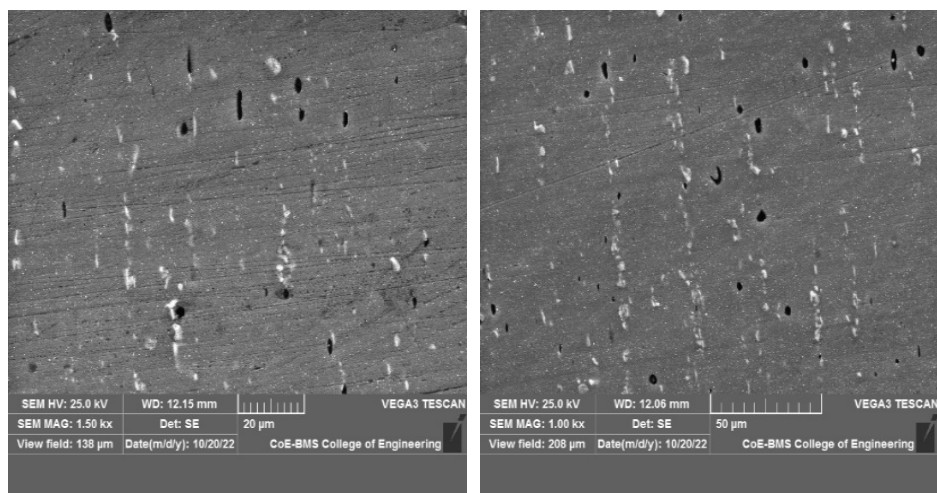

**Figure 2.** SEM images of FSPed AA7075 base metal.

(**a**)

(**b**)

(**c**)

(**d**)

**Figure 3.** *Cont.*

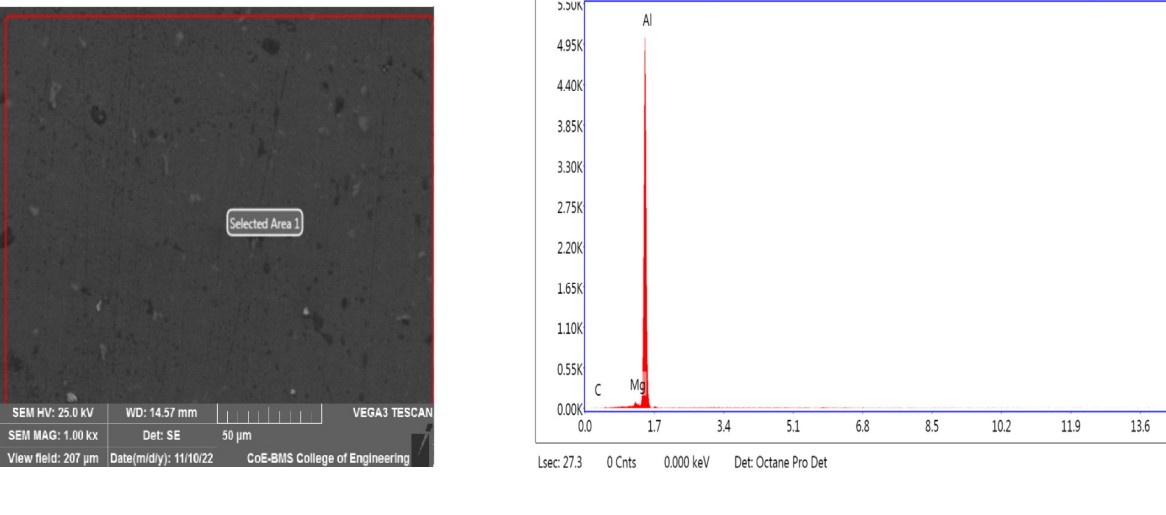

(**e**)                                                            (**f**)

**Figure 3.** XRD analysis of (**a**) selected area and (**b**) EDAX of AA7075/n-B4C; (**c**) selected area and (**d**) EDAX of AA7075/n-TiO$_2$; (**e**) selected area and (**f**) EDAX of AA7075/n-Gr.

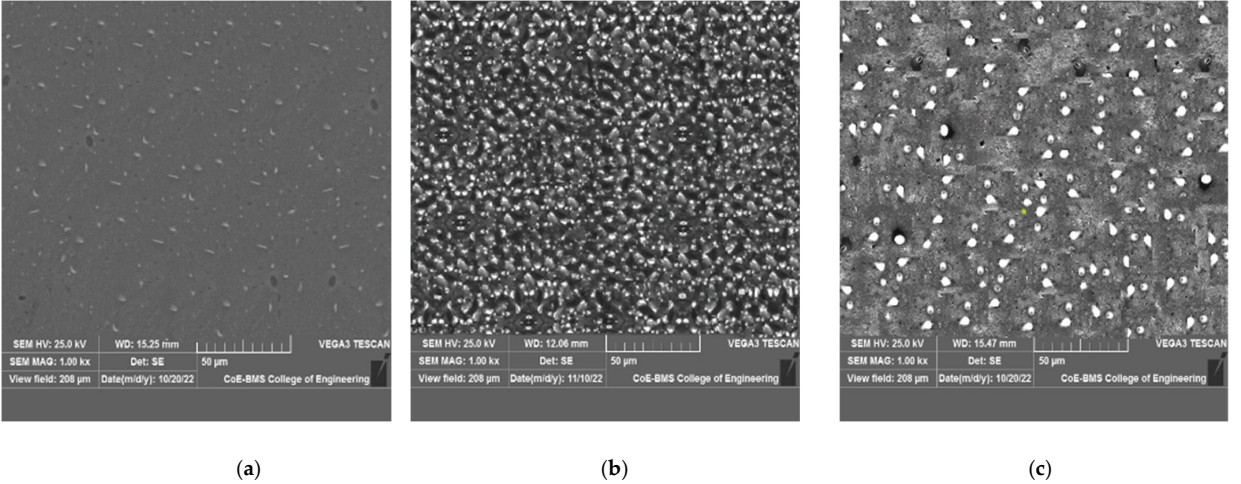

(**a**)                                    (**b**)                                    (**c**)

**Figure 4.** SEM images of a FSPed AA7075/ TiO$_2$ sample prepared at (**a**) 800 rpm and 35 mm/min, (**b**) 1000 rpm and 35 mm/min, and (**c**) 1000 rpm and 45 mm/min.

The stir zone in the treated specimens was distinguished by fine and equiaxed grain structure as a result of plastic distortion and high frictional heat production. It is clear that the FSPed specimen at 1400 rpm exhibited superior particle dispersion, as there was no agglomeration of nanoparticles (TiO2) and finer grain size refinement, which are caused by the production of high heat energy and sufficient time for material movement along the substrate. High rotational speed, which creates novel nucleation sites and reduces grain size, was thought to be responsible for these impacts. To prevent reduced heat intake due to the brief duration of the process, the traverse speed was kept constant at 45 mm/min for various rotation speeds.

SEM images of the FSPed specimens with B$_4$C made at feed rates of 25–45 mm/min and rotating speeds of 800–2000 rpm are shown in Figures 6a–c and 7a–c, respectively. The FSPed specimen's cross-section revealed no flaws or porosities, but there was some aggregation of B$_4$Cparticles, proving that the distribution of the particles in the aluminum matrix was not random. There were relatively few faults apparent in the interface, and it appeared that the surface composite layer was extremely effectively bound to the aluminum alloy AA7075 substrate.

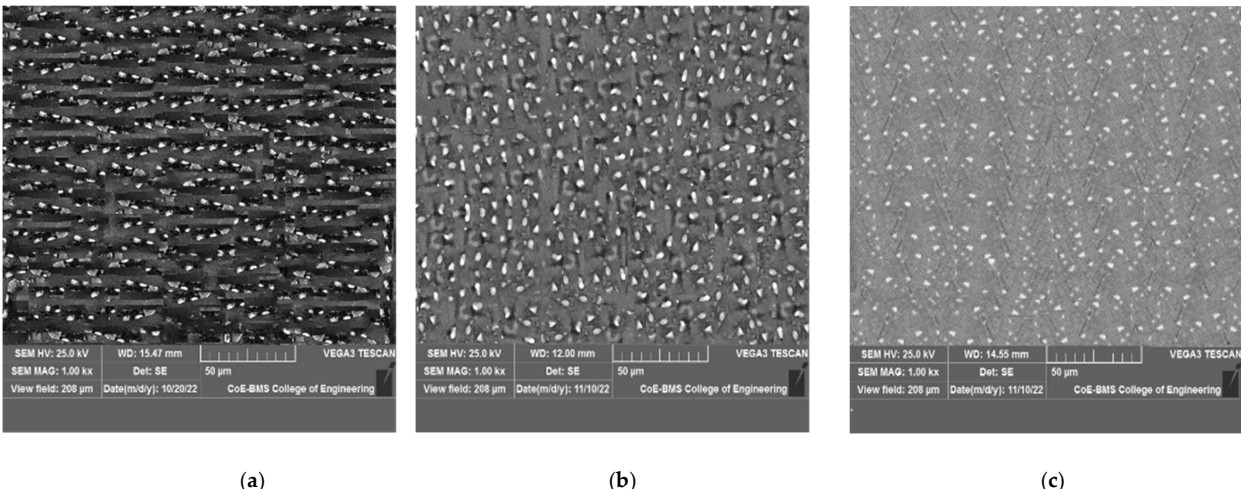

**Figure 5.** SEM images of a FSPed AA7075/ TiO$_2$ sample prepared at (**a**) 1400 rpm and 45 mm/min, (**b**) 2000 rpm and 35 mm/min, and (**c**) 2000 rpm and 45 mm/min.

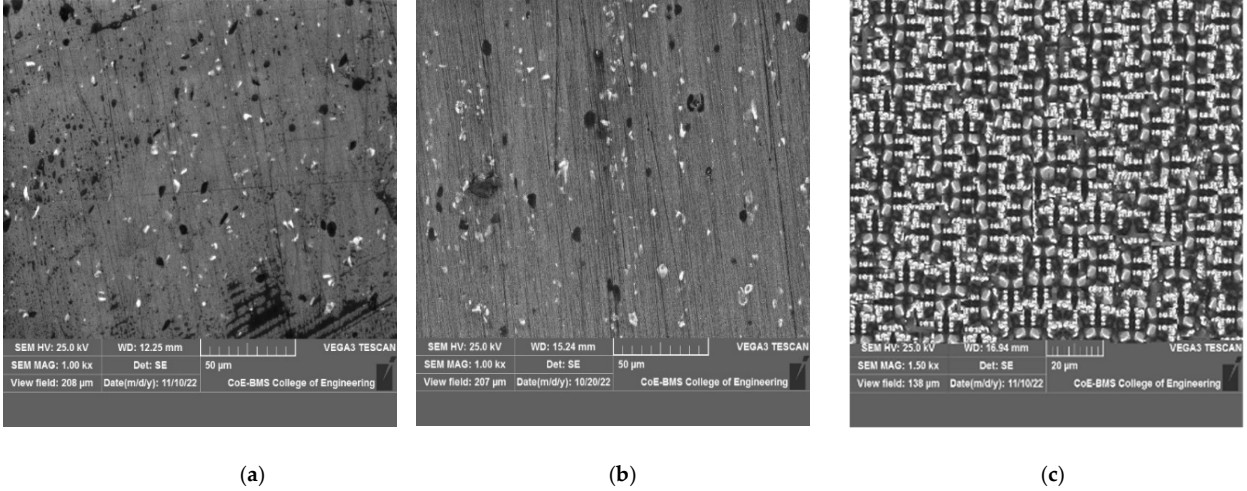

**Figure 6.** SEM images of a FSPed AA7075/ B$_4$C sample prepared at (**a**) 800 rpm and 35 mm/min, (**b**) 41,000 rpm and 45 mm/min, and (**c**) 1400 rpm and 35 mm/min.

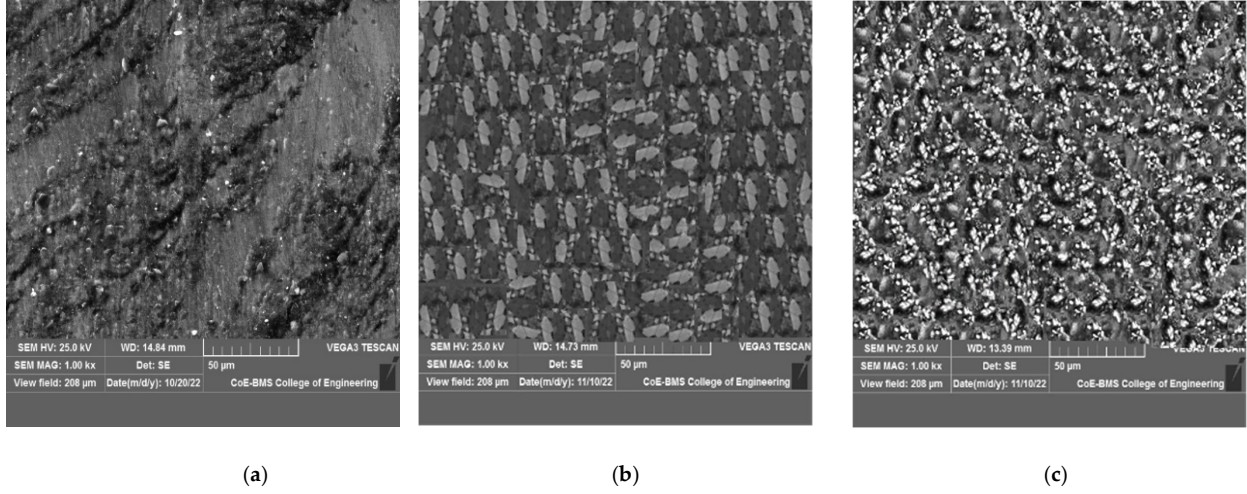

**Figure 7.** SEM images of a FSPed AA7075/ B$_4$C sample prepared at (**a**) 1400 rpm and 45 mm/min, (**b**) 2000 rpm and 25 mm/min, and (**c**) 2000 rpm and 45 mm/min.

No defects or porosities were found in the FSPed specimens generated at the different rotational speeds with a constant traverse speed of 45 mm/min, but there was a small quantity of agglomerated n-B$_4$C particles, demonstrating that the dispersal of particles in the aluminum matrix was not random. The contact between the surface composite layer and the AA7075 substrate seemed to have few defects. Additionally, it is clear from this figure that the particle dispersion in the specimens created using traverse speeds of 45 mm/min and spin speeds of 1400 rpm were superior to those created using 800 and 1000 rpm or greater speeds of 2000 rpm. This leads to the conclusion that the proper mix of tool rotational and traverse rates is essential to achieving a consistent spread of powder particles. Using low rotational speed or high traversal speed does not improve the B$_4$C particle dispersion in the FSP area. The percentage of rotary speed to traversal speed decreases with low rotational velocities or high traverse speeds, which also results in less material stirring.

Figures 8 and 9 display the typical SEM images of the manufactured AA7075/Gr surface composites. These figures show that the Gr particles were not evenly distributed throughout the AA7075/Gr composite, which was produced at 800 rpm rotational speed. In the composites, there was particle accumulation. The typical SEM images of AA7075/Gr surface composites, produced at 1400 and 2000 rpm, indicate that the graphene particles were distributed uniformly throughout the matrix and that they were firmly attached inside them to create a sizable interface.

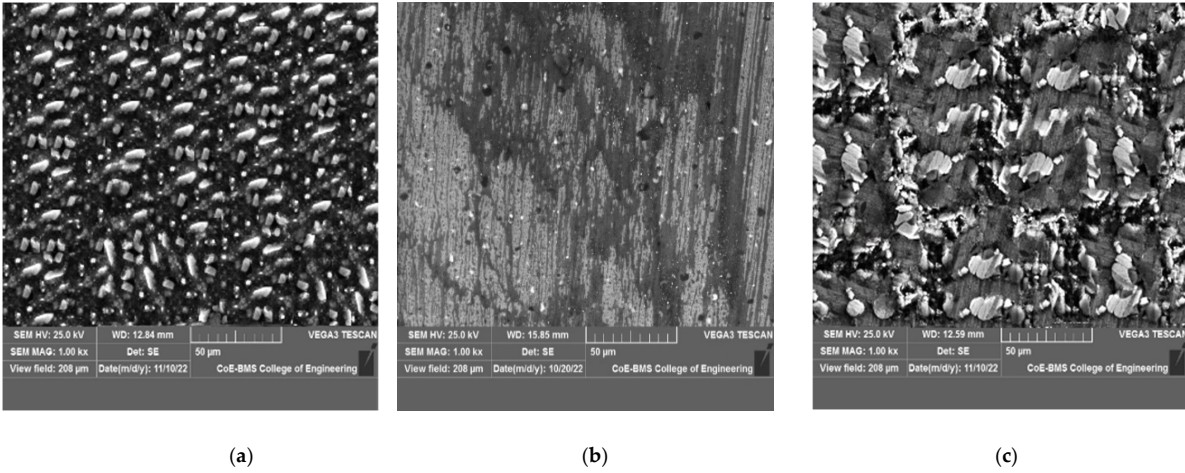

(**a**)        (**b**)        (**c**)

**Figure 8.** SEM images of a FSPed AA7075/ Gr sample prepared at (**a**) 800 rpm and 35 mm/min, (**b**) 1000 rpm and 35 mm/min, and (**c**) 1400 rpm and 25 mm/min.

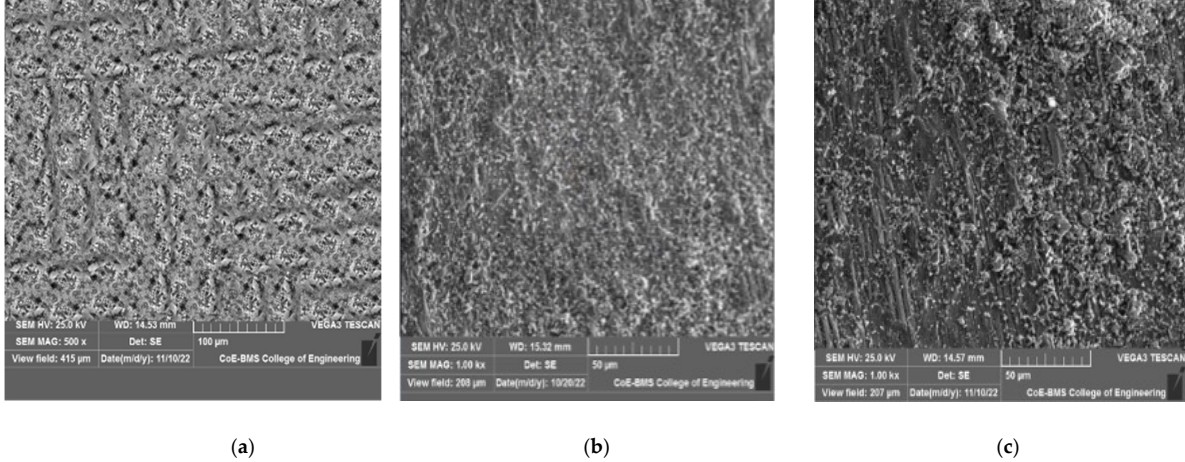

(**a**)        (**b**)        (**c**)

**Figure 9.** SEM images of a FSPed AA7075/ Gr sample prepared at (**a**) 1400 rpm and 45 mm/min, (**b**) 2000 rpm and 35 mm/min, and (**c**) 2000 rpm and 45 mm/min.

A steady traverse speed is desired; it accomplishes excellent bonding with the base substance in order to prevent the pinning effect of the particulates and their aggregation. The banded structure can be eliminated by raising either the tool rotation rate or the rotation rate/traverse speed relationship. Therefore, a greater tool rotation rate increases frictional heating and stirring, which raises the temperature in the stirred zone and results in better material mixing, causing consistent dispersion of nanoparticles.

### 3.2. Micro-Hardness Evaluation

The average value for base metal's hardness was 100 HV, while the average value for an FSPed specimen without particles was 104 HV. Figure 10 shows, at various rotational speeds and feed rates, the micro-hardness profiles of the AA7075/B$_4$C, AA6063/TiO$_2$, and AA7075/Gr surface composites. An excellent dispersion of the graphene particles in the aluminum matrix is also shown by the fact that the micro-hardness of the treated zones is much higher than that of the unprocessed zones when compared to the aluminum matrix AA7075. Even if the particles are evenly distributed throughout the matrix after two FSP passes, the quantity of an upward trend is insufficient when compared to base metal and base metal that has been FSPed.

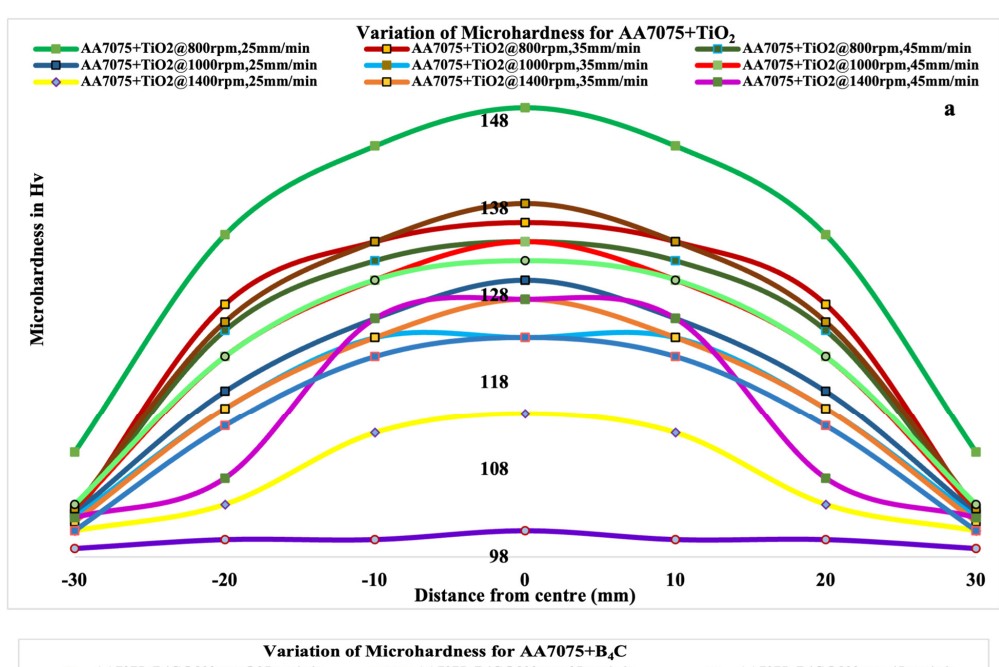

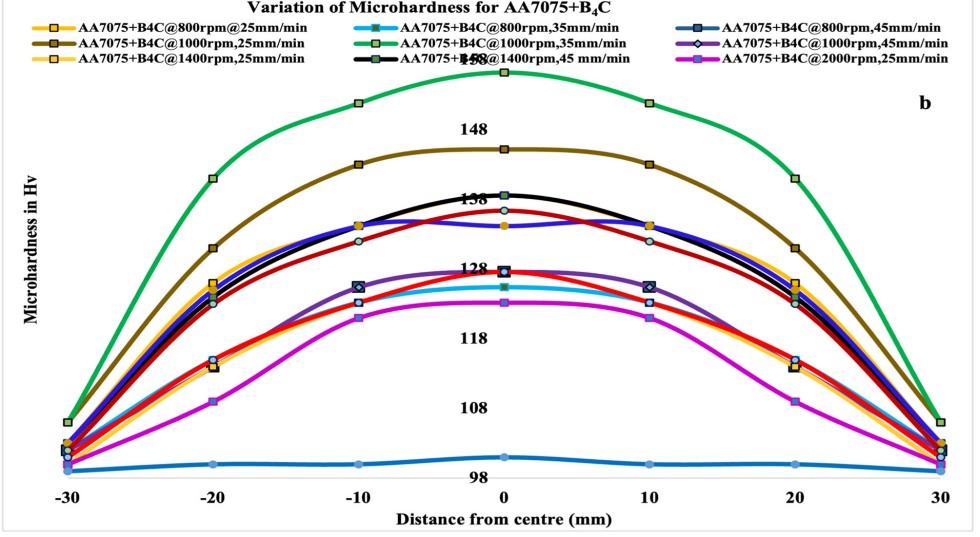

**Figure 10.** *Cont.*

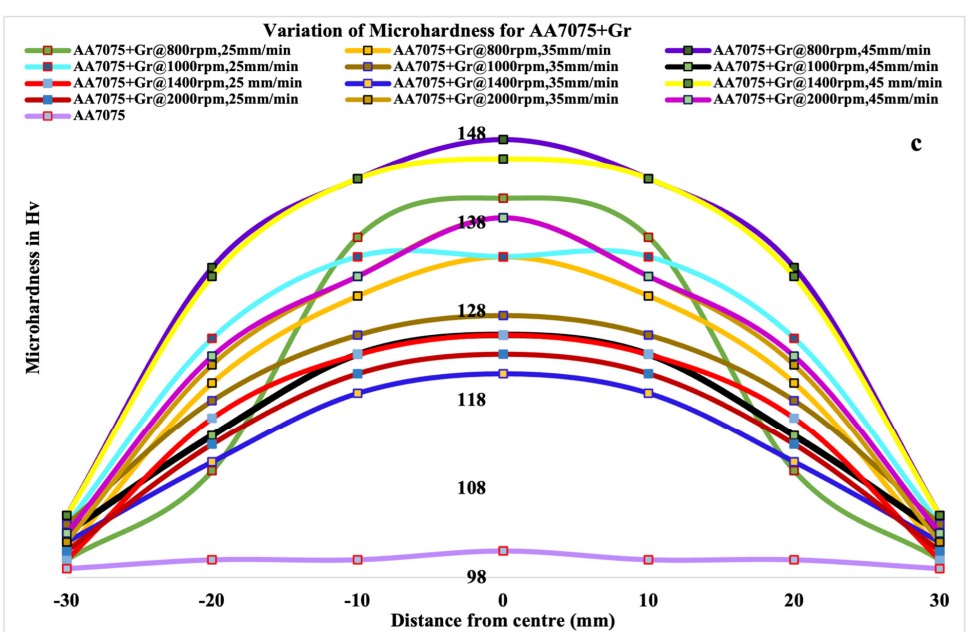

**Figure 10.** Micro-hardness profiles of AA7075 with (**a**) nB$_4$C, (**b**) nTiO$_2$, and (**c**) nGr FSPed.

In comparison to surface composites produced at rotating speeds (1000, 1200, and 1600 rpm), the micro-hardness of AA7075/TiO$_2$ surface composites produced under these conditions revealed a value of 149.6 HV. When compared to the plots for base metal, micro hardness in this instance increased by 48.8% (101 HV). With a feed rate of 35 mm/min and a rotational speed of 1000 rpm, the micro-hardness of the AA7075/B$_4$C surface composite measured 156 HV, which is considerably higher than the values for the surface composites produced at rotational speeds of 800, 1400, and 2000 rpm. As compared to base material, micro-hardness in this instance rose by 54.5% (101 HV).

The micro-hardness A7075/Gr surface composite produced at rotating speeds of 800 rpm and feed rates of 45 mm/min revealed a value of 147.4 HV, which is much greater than that of surface composites produced at rotational speeds of 1000, 1400, and 2000 rpm. In this instance, micro-hardness rose 46% over the figure for base metal (101 HV). The major reason why high hardness values were not obtained as predicted may be due to the soft graphite phase present in all reinforced composites.

When compared to the other two reinforcements, the micro-hardness of the AA7075/B$_4$C surface composite improved the most among the three, which may be attributed to the hardness of the B$_4$C particles. Figure 9 displays the micro-hardness measurements of the FSPed specimens at feed rates of 25–45 mm/min and four different rotational speeds (800, 1000, 1400, and 2000 rpm). The micro hardness levels tend to decrease as the rotating speed increases. This is due to improved dynamic recrystallization caused by the more aggressive stirring of the tool with a pin. Figure 9 also shows that specimens processed at 1400 rpm were harder than specimens processed at 800, 1000, and 2000 rpm. The specimen that was FSPed at 1200 rpm had consistently sized TiO$_2$ particles, and the average particle spacing was quite good.

It has been discovered that as rotating speed grows, so does micro-hardness. This is due to the fact that the increased heat input causes the spinning pin to agitate more vigorously and readily, improving the dispersion of reinforcing particles and increasing the micro-hardness. Low annealing heat input at high rotational speed enhances the micro-hardness behavior. In addition, surface composites' matrix micro hardness levels tend to decrease as rotational speed increases. The matrix softening brought on by the high rotational speed's high heat input, which decreases the micro-hardness, is the root cause of this. The more frequent strengthening precipitates seen in heat-treatable aluminum alloys were made coarser or disintegrated by the matrix softening.

However, because of coarsened precipitates that failed to dissolve during friction stir processing, the thermo-mechanically affected zone (TMAZ) is shown to have a reduction in hardness. Micro-hardness behavior at high rotational speed is enhanced by the low annealing impact of heat input. In addition, the micro-hardness values in the matrix of the surface composites also tend to drop as rotational speed is also increased. This is because high rotating speed and high strength $B_4C$ particles cause rapid input production, which softens the matrix and lowers the micro-hardness.

### 3.3. Surface Roughness

Different profile characteristics, such as Ra (Figure 11a), Rq (Figure 11b), and Rz (Figure 11c), are used to compare the surface roughness of the FSPed AA 7075/$TiO_2$/$B_4C$/ Gr with the FSPed AA 7075. The AA 7075/B4C and AA7075/Gr generated at 800 rpm and 45 mm/min had lower Ra, Rq, and Rz values than the FSPed AA 7075, as shown in Figure 11. Ra, Rq, and Rz values may have decreased as a result of the addition of nanoscale reinforcing particles, which smoothed out the surfaces' tiny peaks and valleys. The non-homogeneous dispersion of nanoparticles may be the cause of the rise in Ra value. The surface roughness parameters (Ra, Rq, and Rz) used to create the brittle surface regions are different from those used in the other three FSPed composite joints in the stir zone, which displayed partly ductile and partially brittle modes. FSPed joint surface characteristics showed that the advancing side TMAZ was next to the nugget region due to the greater temperature and material flow at this side. Consequently, ductile and brittle fracture always result in the object breaking at the weld zone. The failure mode will likely shift from brittle to ductile as the Ra, Rq, or Rz number of FSPed nanocomposites decreases. This shift in failure mode correlates to a change in the elongation. In addition, as FSPed surface parameters (Ra, Rq, and Rz) are low, which result in a decrease in heat generation and a decrease in the quantity of heat transmitted to this region, the width of the heat-affected area of the weld decreases [25]

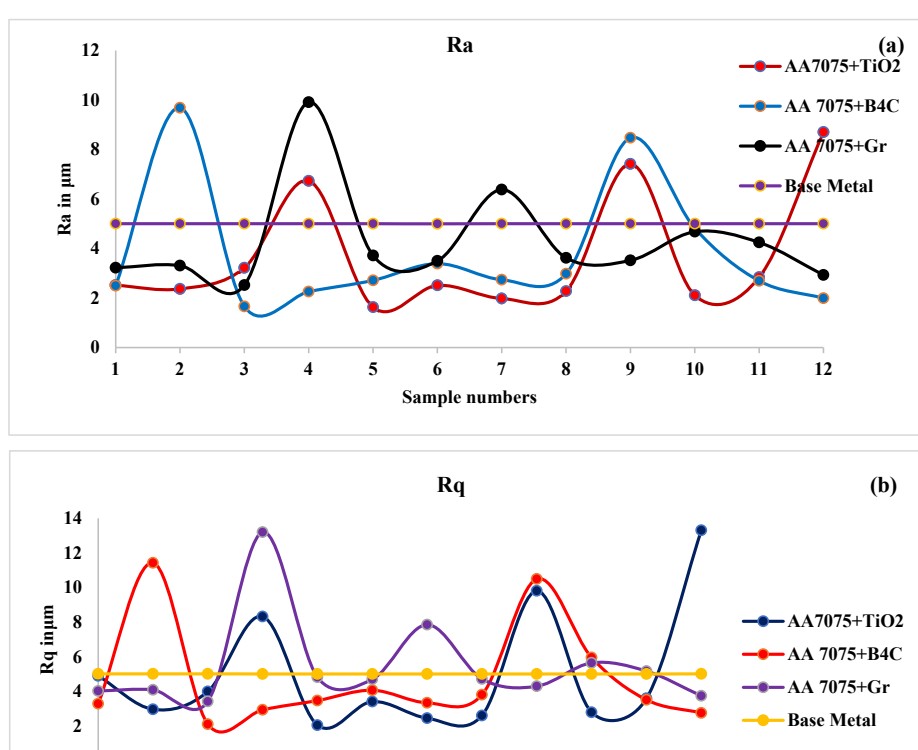

**Figure 11.** *Cont*.

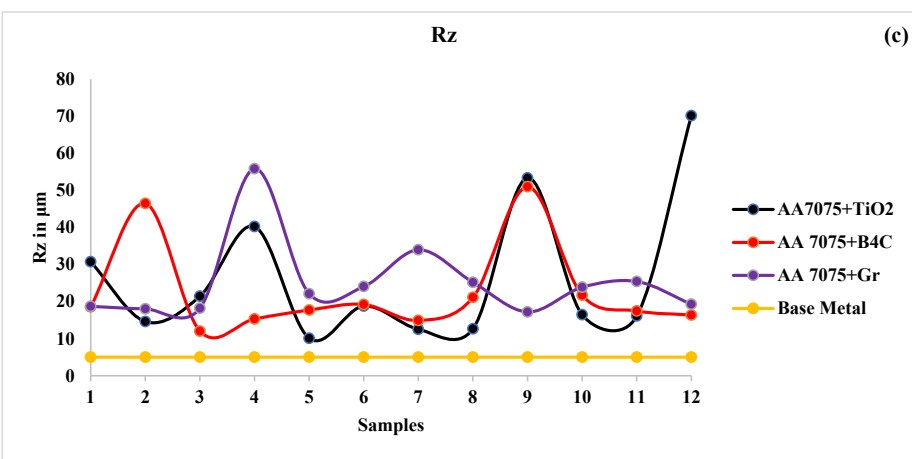

**Figure 11.** Surface roughness in terms of (**a**) Ra, (**b**) Rq and (**c**) Rz of FSPed AA 7075/TiO$_2$/ B$_4$C/Gr produced by selected processing parameters as compared to FSPed AA7075.

## 4. Conclusions

This study examined the impact of different nanoparticles on the micro-hardness, surface roughness and SEM morphology of various surface composites. Results showed that when compared to base material and FSPed samples at 800 and 1400 rpm, the TiO$_2$ and AA 7075 substrate produced at 1400 rpm had satisfactory bonding and had the most homogeneous TiO$_2$ dispersion and the largest grain-reduced size. The micro-hardness values for the surface composite consisting of AA 7075/B$_4$C at 1200 rpm were 101 Hv and 147. 4 Hv, respectively. A micro structural study revealed that homogeneous distribution of particles was achieved at a 1200 rpm rotating speed. The surface micro-hardness was 46% (with Gr), 54.5% (with B$_4$C) and 48% (with TiO$_2$) higher than the substrate material. The produced composite with the highest hardness was the AA7075/B$_4$C surface composite. Thus, hybrid composites may be created to enhance the necessary qualities of aluminum surface composites with the inclusion of transmission electron microscopy investigations. Altering the tool profile allows the use of various methods, such as finite element analysis, for the same investigation.

**Author Contributions:** Conceptualization, M.M.H. and S.Y.; methodology, M.M.H.; software, M.M.H.; validation, M.M.H. and S.Y.; formal analysis, M.M.H.; investigation, M.M.H.; resources, M.M.H.; data curation, M.M.H.; writing—original draft preparation, M.M.H.; writing—review and editing, M.M.H. and S.Y.; visualization, M.M.H.; supervision, M.M.H. and S.Y.; project administration, M.M.H. and S.Y.; funding acquisition, M.M.H. All authors have read and agreed to the published version of the manuscript.

**Funding:** This research received no external funding.

**Data Availability Statement:** Not Applicable.

**Conflicts of Interest:** The authors declare no conflict of interest.

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
