# Peer review of "Investigation on Micro-Hardness, Surface Roughness and SEM of Nano TiO2/B4C/Graphene Reinforced AA 7075 Composites Fabricated by Frictional Stir Processing"

_crystals, doi:10.3390/cryst13030522_

Round 1

Reviewer 1 Report

Authors have presented work on investigation on microhardness, surface roughness and SEM of nano TiO2/ B4C/ Graphene reinforced AA 7075 composites fabricated by frictional stirring processing. Work is interesting, however following comments will be helpful.

References used are not cited in sequence, the in-text numbers need to be adjusted in sequence.

Authors clearly need to present research gaps addressed, thus highlight the novelty and contributions of the presented research. Although the goal is explained in lines 116-117.

Write-up needs to be significantly improved and a connected approach should be followed.

Thoroughly check the manuscript for typos, grammatical errors.

Lines 87-97 are duplicated at lines 98-107, therefore all references must be carefully checked.

Graphical figures are not of good quality and need to be fixed. i.e. Fig. 3 (b,d,f), Fig. 16 and Fig. 17.

The discussion on results could be a lot better.

Conclusion need to be more consolidate and should be in a paragraph form

Author Response

Reviewer 1

  • Authors have presented work on investigation on microhardness, surface roughness and SEM of nano TiO2/ B4C/ Graphene reinforced AA 7075 composites fabricated by frictional stirring processing. Work is interesting, however following comments will be helpful.

References used are not cited in sequence, the in-text numbers need to be adjusted in sequence.

Reply: References are cited in order.

  • Authors clearly need to present research gaps addressed, thus highlight the novelty and contributions of the presented research. Although the goal is explained in lines 116-117.

Reply:  Done.

  • Write-up needs to be significantly improved and a connected approach should be followed.

Reply: carried out

  • Thoroughly check the manuscript for typos, grammatical errors.

Reply: Carried out

  • Lines 87-97 are duplicated at lines 98-107, therefore all references must be carefully checked.

Reply: Duplication is removed.

  • Graphical figures are not of good quality and need to be fixed. i.e. Fig. 3 (b,d,f), Fig. 16 and Fig. 17.

Reply: Figures are modified.

  • The discussion on results could be a lot better.

Reply: Added.

Conclusion need to be more consolidate and should be in a paragraph form

Reply: Consolidated in a paragraph form.

Reviewer 2 Report

The authors investigated the effects of process parameters on the surface quality of the composite made of aluminum alloy 7075 and TiO2, B4C, and graphene. The manuscript is full of grammar errors and awkward sentences. There is even duplicate paragraph and many abbreviations were not well defined. The scientific novelty of the work is very limited. The authors presented many images but the analysis and discussion is insufficient. Although the micro-hardness and surface roughness of the composites were better, the mechanisms of this achievement are still unclear. The sufficient evidences to support the authors ‘claims were not provided. Therefore, I do not recommend the paper for publication.

Author Response

Reviewer 2

The authors investigated the effects of process parameters on the surface quality of the composite made of aluminum alloy 7075 and TiO2, B4C, and graphene. The manuscript is full of grammar errors and awkward sentences. There is even duplicate paragraph and many abbreviations were not well defined. The scientific novelty of the work is very limited. The authors presented many images but the analysis and discussion is insufficient. Although the micro-hardness and surface roughness of the composites were better, the mechanisms of this achievement are still unclear. The sufficient evidences to support the authors ‘claims were not provided. Therefore, I do not recommend the paper for publication.

Reply: Paper has undergone a lot of corrections as per your guidelines and we tried our best to correct it as per your valuable suggestions.

Round 2

Reviewer 1 Report

Authors have addressed reviewer's comments and manuscript has been improved.

Author Response

Thanks for your comments in improving our work.

Reviewer 2 Report

The manuscript has been significantly improved and therefore can be published in Crystals if the following issues can be addressed:

1. There are still some grammar errors in the manuscript that need to be corrected.

2. The author might want to reduce the number of Figures by combining some of the Figures 4-15.

3. The works of https://doi.org/10.1016/B978-0-12-812667-7.00001-X and https://doi.org/10.1016/B978-0-08-102722-6.00006-7 should be cited in the introduction for better review of the applications of carbon-based nanomaterials for nanocomposite application. 

4. Scale bars are required for Figure 1.

5. Figures 3a, c, and, e should have the same scale bar for good comparison.

Author Response

  1. There are still some grammar errors in the manuscript that need to be corrected. Reply: Revised.
  2. The author might want to reduce the number of Figures by combining some of the Figures 4-15.

    Reply: Updated.
  3. The works of https://doi.org/10.1016/B978-0-12-812667-7.00001-X and https://doi.org/10.1016/B978-0-08-102722-6.00006-7 should be cited in the introduction for better review of the applications of carbon-based nanomaterials for nanocomposite application. Reply: Updated.
  4. Scale bars are required for Figure 1. Reply: Updated.
  5. Figures 3a, c, and, e should have the same scale bar for good comparison.  Reply: Updated.